# Multipronged interventions to reduce surgical site infections: A multicenter implementation research protocol

Rachna Rohilla[1]*, Mayank Gupta[2], Thekkumkara Surendran Anish[3], Jerin Jose Cherian[4,5], Mahendra Pratap Singh[6], Ashish Kumar Kakkar[7], Aparna Mukherjee[4], Niti Mittal[8], Sandeep Kaushal[9], Devi Vijay[10], Robin Kaushik[11], Syed Shariq Naeem[12], Jaykaran Charan[13], on behalf of the IMPRESS ('Impact of Multi-Pronged intervention on REducing Surgical Site Infection') Study Group¶

1 Department of Pharmacology, All India Institute of Medical Sciences (AIIMS) Bathinda, Bathinda, Punjab, India, 2 Department of Anaesthesiology, All India Institute of Medical Sciences (AIIMS) Bathinda, Bathinda, Punjab, India, 3 Department of Community Medicine, Government Medical College, Malappuram, Kerala, India, 4 Clinical Studies and Trials Unit, Division of Development Research, Indian Council of Medical Research (ICMR), New Delhi, India, 5 Department of Global Public Health, Karolinska Institute, Stockholm, Sweden, 6 Department of General Surgery, All India Institute of Medical Sciences (AIIMS) Bathinda, Bathinda, Punjab, India, 7 Department of Pharmacology, Postgraduate Institute of Medical Education and Research, (PGIMER), Chandigarh, India, 8 Department of Pharmacology, Postgraduate Institute of Medical Sciences (PGIMS), Rohtak, Haryana, India, 9 Department of Pharmacology, Dayanand Medical College and Hospital (DMC&H), Ludhiana, Punjab, India, 10 Indian Institute of Management, Kolkata, West Bengal, India, 11 Department of General Surgery, Government Medical College & Hospital, Sector-32 (GMCH-32), Chandigarh, India, 12 Department of Pharmacology, Jawaharlal Nehru Medical College (JNMC), AMU, Aligarh, Uttar Pradesh, India, 13 Department of Pharmacology, All India Institute of Medical Sciences (AIIMS), Jodhpur, Rajasthan, India

¶ Other members of IMPRESS Study group are listed in the Acknowledgement section.
* rachna.rohilla20@gmail.com

## Abstract

### Background

Surgical site infections (SSIs) are a major yet preventable cause of poor post-operative clinical outcomes, prolonged ICU/hospital stay, increased antibiotic consumption and added cost of therapy. Low- and Middle-income Countries (LMICs) have disproportionately higher rates of SSIs as compared to high-income countries despite various national and international guidelines in place as multipronged, combined interventions are seldom used. The IMPRESS project aims to respond to this urgent need to identify and evaluate the quality improvement measures contextualized to the logistic constraints of LMIC settings such as India.

### Methods and analysis

We adopt a multi-center longitudinal mixed-methods study to be conducted over a period of 2 years in various phases. Phase 1 will be formative research with the objective of identifying knowledge gaps and baseline data collection. Phase II will involve co-development of multipronged interventions addressing identified barriers. Phase III will focus on the deployment of the selected multipronged interventions. Phase IV will be the

**Data availability statement:** No datasets were generated or analysed during the current study. All relevant data from this study will be made available upon study completion. The study have started baseline data collection only.

**Funding:** This study protocol RR/A5-AM/1.6 version was technically approved by the funding agency and the study will be supported by funding from the Indian Council of Medical Research (ICMR), India under NTF-SRUM file number. ICMR/EM/SRUM/2023/Rachna (E-office No. 171264) dated 06/03/2024; email: icmr.srum@gmail.com. The funder was involved in identifying the research priority, protocol development, and review of the manuscript. The funder will also continue to monitor the study until completion.

**Competing interests:** The funder has appointed the study monitor AK [Clinical Studies and Trials Unit, Division of Development Research, ICMR, New Delhi], under whose guidance the research will be conducted, in coordination with JJC [Clinical Studies and Trials Unit, Division of Development Research, ICMR, New Delhi]. The funder will not be involved in data collection, analysis and interpretation of the results and writing original draft of manuscript. The study monitor and coordinator will however oversee that the timelines are met and study is being conducted as per protocol and following the ethical standards. This does not alter authors' adherence to PLOS ONE policies on sharing data and materials.

post-intervention phase to evaluate the impact of the interventions. The study has been prospectively registered with CTRI and is supported by a funding grant from the Indian Council of Medical Research, New Delhi. The Institutional Ethics Committee approval has been obtained from all the sites involved in the study.

## Article summary

Various interventions have been explored in research for the prevention of SSIs. However, these measures have not been standardized and multipronged combined interventions are seldom used, especially in resource-constrained settings. The strength of our study lies in the design adopted and the semi-structured format, which will allow us to remain open to exploratory findings and help us develop the interventions that are scalable in our healthcare settings. Secondly, our study will be pragmatic. The study deployed over seven sites will enable an understanding of the practical challenges in adopting the interventions and enhance generalizability.

## Introduction

Surgical site infections (SSIs) are defined as superficial site infections, if involving skin and subcutaneous tissue of the incision site within 30 days or deep incisional site infections, if involving deep soft tissues of the incision (fascial and muscle layers) within 30 or 90 days, if implant in situ following the operative procedure [1]. The World Health Organization (WHO) mentions SSI as the most common hospital-acquired infection (HAI) among surgical patients in low-middle income countries (LMICs), affecting around one-third of the patients undergoing surgical procedures [2]. While the global estimates vary from 0.5% to 15% depending on type of surgery, LMICs including India, have disproportionately higher rates of surgical site infections, ranging from 23 to 38% as compared to high-income countries [3,4], despite adjustment for the surgery and patient factors [5]. The SSI rate being higher in emergency procedures as compared to elective procedures [6]. SSIs lead to higher antibiotic consumption, increased medical costs, prolonged hospital stay and recovery time, poor wound healing, risk of wound breakdown and hernia, poor clinical outcomes, psychological challenges and increased mortality [3]. The patients who develop SSI are 60% more likely to be admitted to ICU and have twice the mortality rate than non-infected patients [3]. It is noteworthy that around 40-60% of SSIs are preventable by the use of appropriate infection control practices. Many national and international organizations have published guidelines to prevent SSIs [2,7,8]. Despite these guidance documents in place, SSIs remain a substantial cause of morbidity, mortality and economic burden in hospitalized surgical patients, especially in LMICs. A study from India found a higher prevalence of SSI (17.7%) in emergency surgeries as compared to elective surgeries (12.5%) [9]. Similar rates were found in a recent pragmatic multicentric randomized controlled study in LMICs with an overall SSI rate of 22% (15.5% for clean-contaminated and 30% for contaminated or dirty surgeries) [10]. A recent prospective study from North India found the incidence of SSI to be ~ 8% for clean and ~ 10% for clean-contaminated surgeries [11].

SSI rate may be influenced by factors like pre-operative care, agent and timing of antimicrobial administration for surgical prophylaxis, patient factors such as advanced age and

smoking, co-morbidities especially diabetes, operation theatre environment, intra-operative conditions especially surgery complexity or prolonged surgeries, type of surgery (higher in tumor-related, transplant surgeries), post-operative care and hand hygiene [6,12,13]. However, these measures have not been standardised and multipronged, combined interventions are seldom used. The WHO describes compliance with antibiotic administration within 60 minutes prior to surgery and surgical prophylaxis stopped within 24 hours after surgery as major process measures in reducing the surgical site infections [14]. In addition to the appropriate use of antimicrobial prophylaxis, other measures like maintaining operation theatre (OT) cleanliness, hand hygiene, aseptic precautions, a sterile environment in OT, and restricting the opening of OT doors are other important measures in limiting the SSIs [15,16]. Quality improvement measures in cesarian section patients like following standard antimicrobial prophylaxis guidelines, decreasing OT traffic, reduced door openings, training of OT staff, surgical safety checklist, and biomedical waste management led to reduction in SSI rate from 30% to 5% over a period of 6 months [17]. Continued use of multiple doses of antimicrobial surgical prophylaxis for > 1 day, even in clean surgical procedures [18,19] and use of dual or overlapping antimicrobial coverage (double gram-negative or anaerobic) not only exacerbate antimicrobial resistance but also contribute to the added cost of therapy while increasing the risk of adverse effects [20]. A recent meta-analysis by Cooper et al (2020) showed that education to improve appropriate antibiotic prophylaxis is associated with the reduction of SSIs in LMICs [21].

Despite these advances, the efficacy of optimal surgical prophylaxis has not been strategically evaluated in various surgeries including general surgery, neurosurgery, and pediatric surgery, although some evidence exists for cesarian section. The available evidence emphasises pairing surveillance of antimicrobials with effective audit and feedback mechanisms for frontline workers so that they can modify their behaviour in light of the quantitative data from audit and feedback to improve outcomes [16]. However, this alone has not led to a significant reduction in the rate of SSIs. Hence, there is a need to co-develop feasible and scalable multipronged interventions targeting healthcare professionals for the prevention of surgical site infections. These interventions must address the system-level challenges and be contextualised to the logistical constraints of LMICs.

## Study objectives

The primary objective of the IMPRESS study is to evaluate the effect of multipronged interventions on the rate of surgical site infections in clean, clean-contaminated and contaminated surgeries.

The secondary objectives (SO) of the study are:

1) to identify knowledge gaps in the practice, barriers and challenges in the adoption of surgical prophylaxis prevention guidance;

2) to co-develop multipronged interventions that can be applied in hospital settings in LMICs;

3) to evaluate the impact of multipronged interventions on in-patient mortality and length of hospital stay;

4) to evaluate the number of ICU admissions averted and number of hospital readmissions averted within 30 days after discharge by adoption of multipronged interventions;

5) to evaluate the effect of multipronged interventions on the antimicrobial consumption indicators Days of Therapy (DOT);

6) to evaluate the percentage of patients receiving the correct surgical antimicrobial prophylaxis.

Fig 1 map the secondary objectives to the specific research questions.

## Methods and analysis

### Study design and study process

A quasi-experiment (pre-post design) will be adopted for the study. Our research design adopts a multi-sited mixed-methods approach across a four- phased strategy, each phase addressing a specific research question (Figs 1, 2).

Phase I (Formative research). Phase I will involve a scoping review with the objective to identify the knowledge, attitude and practice (KAP) gap in the adoption of surgical prophylaxis prevention guidance in the LMIC setting. In addition to the scoping review, a mixed-method analysis in the form of a sequential exploratory design, and cross-sectional analysis of administrative data (quantitative) followed by in-depth interviews (IDIs) of surgeons and nurses, as well as observations (qualitative) will be conducted to achieve the first secondary objective (SO-1).

The quantitative analysis involves the baseline data collection on the burden of SSI and the evaluation of surgical practices among healthcare professionals using a cross-sectional analysis. The risk factors which might contribute to the increased risk of SSI will be identified. The qualitative semi-structured questionnaire will be developed from the scoping review.

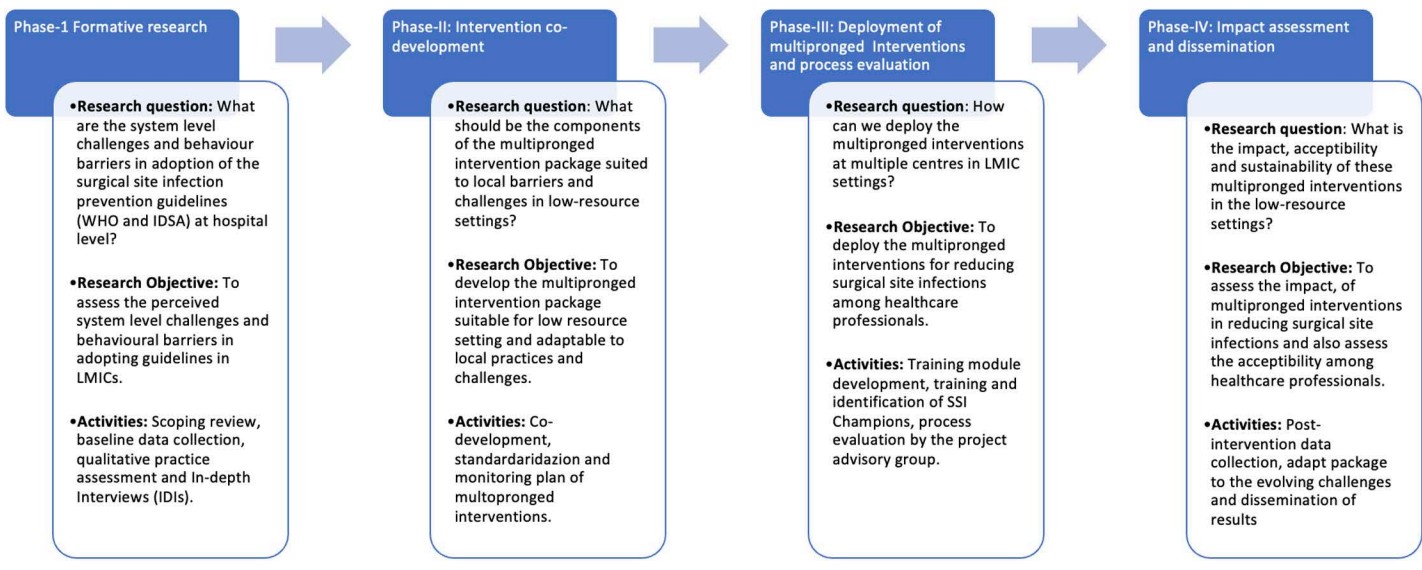

**Fig 1. Phase-wise research strategy, objectives and activities planned for each phase of the study.**

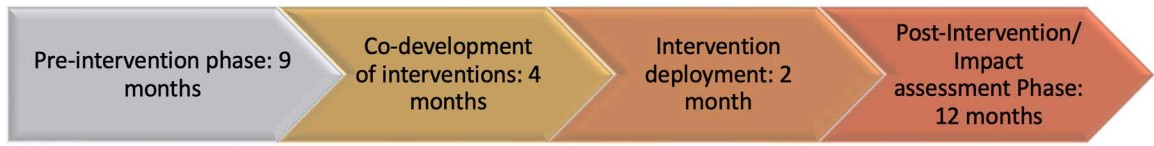

**Fig 2. Study plan and duration of each phase of the study.**

The questionnaire will be filled by the surgeons, nurses and other healthcare professionals in direct contact with the patient, and the questions will explore the practice followed in their setting (elective and emergency surgeries) and if there are any challenges in the adoption of the already published guidelines (global guidelines for prevention of surgical site infections by WHO second edition 2018 and clinical practice guidelines for antimicrobial prophylaxis in surgery) [2,7]. Overall, the qualitative component allows us to explore surgeons' perspectives and perceived barriers to the implementation of SSI prevention strategies in their respective settings. A semi-structured format allows us to remain open to exploratory findings during the course of interviews. The number of interviews will range from 10-15 per site or until saturation of data is achieved.

The baseline data collection will be done by the trained research associate/staff nurse for the eligible patients undergoing surgeries and the baseline SSI rate in the setting will be calculated. The baseline data collection will be done using the standard case record form (CRF) designed for the study which will include the parameters (but not limited to) like type of surgery, American Society of Anesthesiologists (ASA) physical status score, timing and antimicrobial agent for surgical prophylaxis, duration of surgery, redosing of antimicrobial prophylaxis, and other defined parameters.

In addition, the trained research associate/staff nurse will observe the practices of the care delivery team and observe interactions between the care team and patients. Detailed observation notes will be recorded on a daily basis, and summarized across sites on a fortnightly basis. This multi-sited collaborative approach will enable the team to attend to practices that may be observed at one site and not the other, thus forming a shared repository. This qualitative assessment will allow the research team to observe points that practitioners may not have reflected on or recollect during interviews. The baseline data collection phase will last for 6 months.

Outcome of Phase I: The expected outcome of Phase I will be to identify the knowledge gaps in the practice, barriers and challenges in the adoption of surgical prophylaxis guidance (SO-1). The baseline data collection will be done in this phase (pre-intervention data) on the baseline SSI rates and practices followed in the setting. These findings will be helpful in the co-development of the multipronged interventions during Phase-II.

Phase II (Co-development of a multipronged intervention package by the stakeholders addressing the identified barriers and suitable for low resource settings). The multipronged interventions will be co-developed by the stakeholders based on the findings of the Phase-I, including the scoping/literature review. The stakeholders will involve the head of the departments of surgery, hospital administration (medical superintendent of the site), and the investigators (PI and Co-PI) from the respective sites involved. The most relevant, feasible and impactful interventions will be identified from Phase I.

A series of discussions on evidence-based interventions and foreseen challenges with all the stakeholders will be conducted to finalize the components of the multipronged intervention. These interventions will then be first be listed, mapped, and all the stakeholders will reach a consensus on components to be included as the multipronged intervention.

After developing the multipronged interventions by the stakeholders, the review will be done by the Project Advisory Group (PAG). The PAG will involve senior academicians, external experts in the implementation research and PIs from all the sites. The final multipronged interventions will cover various core domains - system, structural and behavioral; which will be implemented in Phase III.

After co-development, the standardization of these multipronged interventions will be done and monitoring plan for the same will be drawn.

Intervention standardization It includes defining the type of standardization for each component and defining the limits of the standardization for each component [22,23]. The

type and limits of the standardization will be set after consensus with the stakeholders and the project advisory group.

A.  The type of standardization for each component and step will be classified into "mandatory", "optional", or "prohibited". The "mandatory" step must be performed under all circumstances and, if not performed, will be considered as protocol deviation; the opposite is true for "prohibited". The "optional" step may or may not be performed at the discretion of the participating surgeon [23].

B.  The limits of standardization for each component will be set as "exactly", "with boundaries" or "without boundaries (flexible)" based on the flexibility allowed for the adoption by the health care professional. If mentioned as "with boundaries", the component or step can be followed as per the amount of flexibility mentioned for each component of the intervention [23].

**Intervention monitoring** The levels of fidelity will be defined and noted as per the discussion with all the stakeholders. Any discrepancy among the stakeholders will be resolved after discussion with each other to reach a final consensus. Based on the multipronged interventions, a checklist and standard data collection form will be developed. The checklist developed will be used for assessing the level of fidelity and compliance to the multipronged intervention package for each patient.

Outcome of Phase II: Feasible and acceptable multipronged intervention package that can be implemented in the low-resource setting to prevent surgical site infections.

Phase III (Deployment of the multipronged interventions).  Multipronged interventions developed in Phase II will be implemented in Phase III as a quasi-experimental study using the following strategies: training of healthcare professionals, identification of surgical site infection prevention (SSIP)/stewardship champions, review of implementation by the project advisory group. The primary objective of the study and secondary objectives 3-6 will be accomplished in this phase.

A. **Development of a structured training curriculum and training manual based on the intervention package**: The structured training modules will be developed as per the multipronged interventions co-developed in Phase II. The training modules will be contextualized to the practice change needs and the training sessions will be conducted using interactive sessions and practical demonstrations. The assessment will be applied pre and post-training among the health care professionals undergoing training to assess the practice and knowledge change adopted from the training modules. The assessment will also indicate the need for re-training.

B. **Training and skill development:** The project steering group (comprising of a multidisciplinary team of PIs from the sites, and antimicrobial stewardship experts) will impart comprehensive training to the relevant healthcare professionals including the operation theatre nursing staff involved in surgery and patient care, the residents, anesthesiologists and the surgeons using the structured training modules and curriculum using 4-5 interactive sessions, didactic lectures for adoption of the multimodal interventions spanning over a period of two months. The healthcare professionals undergoing training will be assessed for practice and knowledge change using a pre-developed assessment module. A score of > 80% in the training assessment will be considered adequate. If the score is < 80%, then a re-training session will be conducted.

C. **Implementation of the multipronged intervention package:** The health care professionals will implement the multipronged interventions in toto after completion

of necessary training. The impact on SSI and other secondary objectives will then be recorded. Any challenge faced in the adoption of the interventions will be documented and communicated.

D. **Identification of Surgical Site Infection Prevention (SSIP)/Stewardship Champions:** The "SSIP champions" will be identified for each setting by the site PI/Co-PI (could be ICN nurse/residents/surgeons who have leadership qualities and are actively working in the surgical department). These SSIP champions will ensure the adoption of multimodal interventions in their setting and will also assess the need for interim training or re-training in their respective sites. If the compliance with the multipronged interventions is lacking at a particular site as assessed by the checklist of each patient, then re-training will be scheduled. If non-compliance is persistent, then it will be communicated to the project advisory group. The process evaluation will be done by the project advisory group, and IDI will be conducted to understand the hurdles and challenges in the adoption of these interventions, as explained below.

Process evaluation: The compliance to the multipronged interventions, as reported by the SSIP champions, will be evaluated by the project advisory group. The SSIP champions will also evaluate whether the structural and system changes are in place at respective sites and whether behavioural changes are adopted by the health care professionals. If the non-compliance is persistent, then the stakeholders will re-assess the acceptance of the co-developed multipronged interventions. In-depth interviews with the health care professionals will follow to understand the hurdles and challenges in accepting these interventions and troubleshooting iteratively. The multipronged interventions will be either adapted as per the challenges noted or co-developed as per the site need or if only retraining will suffice, then re-training will be planned. The monitoring of compliance to the multipronged interventions at each site using the checklist will be done by SSIP champions and reported to the project advisory group quarterly.

The intervention will continue with the help of SSIP Champions. The intervention and quarterly result assessment will go hand in hand and the impact assessment will follow.

Phase IV (Post-intervention Phase). This phase will involve the data collection on the same parameters as in the pre-intervention phase using the CRF to evaluate the implementation of multipronged intervention packages using the developed checklist. The SSIP champions will evaluate whether these multipronged interventions are adopted adequately in their respective setting. Intermittent weekly reminders such as by e-posters will be sent to health care professionals for adoption of multipronged interventions, which will act as reinforcement reminders. Interim training sessions will be planned every 6 months (in case new staff is recruited or diminished compliance is noted).

In-depth interviews will be conducted with key stakeholders (surgeons, health care professionals and nursing officials involved) to evaluate their perception of impact and feasibility of interventions at 6 months post-intervention phase and suggestions to improve the multipronged interventions.

The study period and timelines of each phase of the study have been shown in Fig 3

## Ethical approval and consent to participate

The study will follow all the principles laid down by the Declaration of Helsinki. The Ethical Clearance has been obtained from the Institutional Research Board (IRB) and Institutional Ethics Committee (IEC) from the Central coordinating site (AIIMS Bathinda) vide approval number IEC/AIIMS/BTI/554 dated 13-04-2024 and each participating site by respective Site PIs. The consent has been taken from the healthcare professionals for participation in the

study as the intervention is directed towards healthcare professionals. We obtained waiver of informed consent for participants from IEC as the interventions are directed towards healthcare professionals and patients will be followed only for outcomes in anonymized form. The multipronged intervention will direct towards healthcare professionals (system or administrative or behavioral). The patient data will be kept confidential and will be recorded as anonymized form during and after the study. Due approval has been taken from competent authority and the surgeon in-charge for the accessing the health records of the patients.

The informed consent will be obtained from healthcare professionals before the In-depth Interviews (IDIs). The qualitative data will be kept confidential. The results will be presented and published in anonymized form.

The study has been prospectively registered with CTRI with registration number CTRI/2024/04/066362 dated 26th April 2024. Any amendments/protocol modifications, if required will be communicated to the IEC at each site and necessary approvals will be obtained.

## Current status of recruitment of patients for Phase-I of the study

The baseline data collection as per Phase-I has started from 1st September 2024. The duration of study will be 3 years.

**Study duration, enrollment and number of sites.** The study participation for patients will last for 30 days unless an implant is placed during surgery. Each subject will be included in the study on pre-operative day 1 and followed till discharge. After discharge, the patient will be reviewed in the OPD or followed up telephonically till 30 days after the surgery or 90 days if the implant is placed (where the day of surgery will be taken as POD-0) for any evidence of

| SPIRIT FIGURE | STUDY PERIOD | | | | | |
|---|---|---|---|---|---|---|
| PHASES OF THE STUDY | PHASE-I | PHASE-II | PHASE-III | PHASE-IV | Close-out | Publication and Dissemination |
| TIME FRAME | 0 | t1 | t2 | t3 | t4 | t5 |
| DURATION (IN MONTHS) | 9 | 4 | 2 | 12 | 2 | 2 |
| ELIGIBILITY SCREEN | ✓ | | | ✓ | | |
| PRE-INTERVENTION DATA COLLECTION | ✓* | | | | | |
| DESIGNING MULTIPRONGED INTERVENTIONS | | ←→ | | | | |
| INTERVENTION PHASE | | | ←→ | | | |
| POST-INTERVENTION DATA COLLECTION | | | | ←→ | | |

*The baseline data collection at all the sites, scoping review and literature review has been started and is ongoing.

**Fig 3. SPIRIT figure showing various phases of study along with the proposed time frame.**

surgical site infection. Any re-admission in between the discharge and 30 days will be noted along with the reasons for re-admission.

There will be seven collaborating sites across different Indian states with the central coordinating site being All India Institute of Medical Sciences (AIIMS) Bathinda, Punjab, India. Other six are as follows: Postgraduate Institute of Medical Education and Research, (PGIMER), Chandigarh, India; Postgraduate Institute of Medical Sciences (PGIMS), Rohtak, India; Dayanand Medical College and Hospital (DMC&H), Ludhiana, India; Government Medical College & Hospital, Sector-32 (GMCH-32), Chandigarh, India; Jawaharlal Nehru Medical College (JNMC), AMU, Uttar Pradesh, India; All India Institute of Medical Sciences (AIIMS), Jodhpur, Rajasthan, India.

The multi-center collaborative approach presents novel possibilities for developing practical and generalizable toolkit.

**Study population.** Surgical patients undergoing elective or emergency procedures (open or laparoscopic) under general surgery, neurosurgery, orthopedics, pediatric surgery, plastic surgery or urology will be included in the study. Immunocompromised patients including uncontrolled diabetic patients, cancer patients, on immunosuppressive therapy; patients with HIV/HBV/HCV infection; patients with prolonged hospital stay > 1 week and receiving multiple antimicrobials before the planned procedure; patients with dirty wounds or pre-existing active infection at surgical site; patients requiring therapeutic hypothermia during intra-operative or post-operative period, patients with American Society of Anesthesiologist (ASA) score [24] of 6 (declared brain dead whose organs are being removed for donor purposes) will be excluded. In addition, minor procedures performed under local anesthesia; pregnant or lactating females will be excluded from the study.

**Sample size and power.** The three strata based on the type of the surgery (clean, clean-contaminated and contaminated) will be separately powered based on different baseline SSI rates extracted from an Indian study showing the prevalence of SSIs in general surgery patients [8,9]. The sample sizes were based on 90% power, a 5% two-sided significance level and 15% loss to follow-up or death before reaching the primary endpoint at 30 days. For the clean surgeries, anticipating the baseline average SSI rate of 8%, a 3% absolute reduction in SSI to 5% will be taken as clinically significant and would require 3260 patients in total (1630 patients in pre-intervention and intervention each). For the clean-contaminated surgeries, anticipating the baseline average SSI rate of 12%, a 4% absolute reduction in SSI to 8% will be taken as clinically significant (i.e., relative risk of 0.67) and would require 2700 patients in total (1350 in pre-intervention and intervention each). For the contaminated surgeries, anticipating higher baseline SSI rate of 30%, a 10% absolute reduction in SSI to 20% will be taken as clinically significant and would require 900 patients in total (450 in pre-intervention and intervention each).

**Data collection and management.** The patient planned for elective surgery will be screened as per inclusion-exclusion criteria a day prior to the surgery. For emergency surgeries, the screening will be done as earliest possible before planned emergency surgery.

The anonymized data will be collected in the standard data collection form. The confidentiality of the data will be maintained during, after the study and during publication of the results.

The data collection for the surgical patients undergoing elective or emergency surgery will be done by the nurse and research fellow in the standardized data collection form. The patient will be followed up regularly till discharge, and data on SSI, antimicrobials, and clinical outcomes will be recorded using RedCap database and source documents will be stored with patient wise folders at each site. The patient will be called in person for follow-up to ODP at

30 days to know the clinical status and outcome of the patient. Any unplanned visit to OPD or emergency will be noted with the reason for the visit.

The in-depth interviews (IDI) records will be stored to ensure data quality at the central coordinating site and later will be analyzed as per data analysis plan.

In the intervention phase, in addition, the checklist of multipronged interventions will be checked for implementation assurance along with the data collection form.

**Subject completion/withdrawal.** The patients who take leave from the hospital against medical advice (LAMA) will be contacted for follow-up at 30 days for the outcome.

If follow-up is not possible at 30 days after surgery, patients will be followed up as soon after this as possible or telephonically. If patients develop SSI before postoperative day 30, they will still be reviewed at 30 days after surgery to record secondary outcomes.

**Study outcomes.** The primary outcome of the study is the rate of surgical site infection till 30 days after surgery (90 days if implant) using CDC definition of SSI in pre-intervention versus post-intervention phase for clean, clean-contaminated and contaminated surgeries

The secondary outcomes are (1) the number of ICU admissions averted and number of readmissions averted in clean, clean-contaminated and contaminated surgeries by adoption of multipronged interventions; (2) the length of index hospital admission in pre-intervention and post-intervention phase for clean, clean-contaminated and contaminated surgeries; (3) the antimicrobial consumption indicators (Days of therapy) in the pre-intervention and post-intervention phase for clean, clean-contaminated and contaminated surgeries; (4) the percentage of patients receiving single dose of antimicrobial prophylaxis in the pre-intervention and post-intervention phase for clean, clean-contaminated and contaminated surgeries; (5) the percentage of patients receiving prolonged antimicrobial prophylaxis (>24 hours duration) in the pre-intervention and post-intervention phase for clean, clean-contaminated and contaminated surgeries; (6) the percentage of irrational combinations of drugs (double gram positive, double gram negative or double anaerobic coverage) for surgical antimicrobial prophylaxis in the pre-intervention and post-intervention phase for clean, clean-contaminated and contaminated surgeries; (7) the number of deaths prevented in clean, clean-contaminated and contaminated surgeries by adoption of multipronged interventions; and (8) acceptance of these multipronged interventions and challenges in their implementation by healthcare professionals.

**Statistical analysis plan.** The data collected using the standard data collection form will be analysed after entry into the excel form. The antimicrobial consumption analysis will be done using Microsoft Excel. The descriptive data analysis will be done using Statistical Package for the Social Sciences (SPSS). The rate of surgical site infection till 30 days after surgery (90 days if implant) will be expressed as events per person-time, separately for clean, clean-contaminated, contaminated surgeries. All secondary endpoints will be expressed as mean, standard deviation (if the variable is continuous) or as frequency, percentage (if the variable is categorical). The rates of primary outcome and secondary outcomes measured at the study centres in post-intervention will be compared with the pre-intervention/baseline data to assess the impact of the intervention. Chi-square test, independent sample t-test or Mann-Whitney U test will be used for the statistical comparisons.

Plots and pivot tables will be generated using statistical software or excel as appropriate. The analysis will be conducted separately for clean, clean-contaminated and for contaminated surgeries. A sub-group analysis will also be conducted as per the level of fidelity.

Kaplan-Meier curves will be plotted to demonstrate the incidence rates of surgical site infections. The difference in the rate of surgical site infections between the nature of surgeries (clean, clean-contaminated, contaminated surgeries), or any other exposure categories will be

statistically tested using the log-rank test. A Cox proportional hazard model will be constructed to identify the predictors of surgical site infections.

**Qualitative methods and analysis.** We will adopt qualitative research approaches in two phases of our study. In-depth interviews will be conducted in phase 1 (formative study) as the design of the same is a mixed-method, sequential exploratory design. As mentioned earlier, interviews will also be conducted with key stakeholders (surgeons, nursing officials and other health care professionals involved) to evaluate their perception of the impact and feasibility of interventions at 6 months post-intervention phase and suggestions to improve the multipronged interventions (Phase 4). Thematic analysis will be the analytical framework used for the qualitative analysis [25].

All interviews will be transcribed in English. The PI and Co-PIs will read through the responses in the qualitative practice assessment, observation notes and interview transcripts, creating memos where necessary. All responses will be thematically coded to facilitate the identification and analysis of patterns or themes in the data set [25,26]. Themes will be generated from the initial codes. Themes will be reviewed, named and documented. Multiple thematic codes will be shared with all members of the research team to guide the development of the multipronged interventions.

**Dissemination.** The project advisory group (PAG) will maintain the audit trail of all the activities done in their respective sites. Periodic meetings will be conducted (physical or virtual with all the stakeholders and they will be appraised of the project activities). The implementation plan will be continuously evolved with the inputs for the project advisory group and external experts in implementation research. The report of the project will be presented to the Heads of the respective institutes and the PAG. The results will be published in an indexed journal with a reputable standing in the research community. The evidence generated from this study will be used to advocate for prevention of SSIs in hospital settings, particularly in the Global South, but also of potential interest in resource-constrained settings in different parts of the world today.

## Supporting information

**S1 File. IEC AIIMS Bathinda approved protocol.**
(DOCX)

**S2 File. Gantt chart.**
(XLSX)

**S3 File. SPIRIT checklist.**
(PDF)

## Acknowledgment

We would like to acknowledge the contributions of the members of the Central Coordinating Unit (CCU) team, Technical Advisory Group (TAG) and the Indian Council of Medical Research (ICMR) -National Task Force (NTF) for Safe and Rational Use of Medicine (SRUM). Their support in protocol development and data management plan are greatly appreciated. We are thankful to Dr. Lokesh Sharma, Scientist at ICMR-NIOH, Ahmedabad, Gujarat for agreeing to provide us the guidance and support for data management using ReDCap.

The lead author of the **IMPRESS ('Impact of Multi Pronged intervention on REducing Surgical Site Infection')** Study Group is Rachna Rohilla, Department of Pharmacology, All India Institute of Medical Sciences (AIIMS) Bathinda, Punjab, India (email: rachna.rohilla20@gmail.com) who is the Principal Investigator of the study.

Other members of the IMPRESS Study Group are listed below:

- Mayank Gupta, Department of Anaesthesiology, All India Institute of Medical Sciences (AIIMS) Bathinda, Punjab, India
- Thekkumkara Surendran Anish, Department of Community Medicine, Government Medical College, Manjeri, Malappuram, Kerala, India
- Jerin Jose Cherian, Clinical Studies and Trials Unit, Division of Development Research, Indian Council of Medical Research (ICMR), New Delhi, India
- Mahendra Pratap Singh, Department of General Surgery, All India Institute of Medical Sciences (AIIMS) Bathinda, Punjab, India
- Ashish Kumar Kakkar, Department of Pharmacology, Postgraduate Institute of Medical Education and Research, (PGIMER), Chandigarh, India
- Aparna Mukherjee, Clinical Studies and Trials Unit, Division of Development Research, Indian Council of Medical Research (ICMR), New Delhi, India
- Niti Mittal, Department of Pharmacology, Postgraduate Institute of Medical Sciences (PGIMS), Rohtak, India
- Sandeep Kaushal, Department of Pharmacology, Dayanand Medical College and Hospital (DMC&H), Ludhiana, India
- Devi Vijay, Indian Institute of Management, Calcutta, Kolkata, India.
- Robin Kaushik, Department of General Surgery, Government Medical College & Hospital, Sector-32 (GMCH-32), Chandigarh, India
- Syed Shariq Naeem, Department of Pharmacology, Jawaharlal Nehru Medical College (JNMC), AMU, Uttar Pradesh, India
- Jaykaran Charan, Department of Pharmacology, All India Institute of Medical Sciences (AIIMS), Jodhpur, Rajasthan, India
- Simrandeep Singh, Department of General Surgery, Government Medical College & Hospital, Sector-32 (GMCH-32), Chandigarh, India
- Yashwant R Sakaray, Department of General Surgery, Postgraduate Institute of Medical Education and Research, (PGIMER), Chandigarh, India
- Sanjay Marwah, Department of General Surgery, Postgraduate Institute of Medical Sciences (PGIMS), Rohtak, India
- Jaspal Singh, Department of General Surgery, Dayanand Medical College and Hospital (DMC&H), Ludhiana, India
- Shahbaz Habib Faridi, Department of General Surgery, Jawaharlal Nehru Medical College (JNMC), AMU, Uttar Pradesh, India
- Mohammad Jesan Khan, Department of Orthopedics, Jawaharlal Nehru Medical College (JNMC), AMU, Uttar Pradesh, India
- Naveen Sharma, Department of General Surgery, All India Institute of Medical Sciences (AIIMS), Jodhpur, Rajasthan, India
- Abhay Elhence, Department of Orthopedics, All India Institute of Medical Sciences (AIIMS), Jodhpur, Rajasthan, India

The patients or the public were not involved in designing the protocol of the study.

## Author contributions

**Conceptualization:** Rachna Rohilla.

**Funding acquisition:** Rachna Rohilla, Mayank Gupta, Jerin Jose Cherian.

**Methodology:** Rachna Rohilla, Mayank Gupta, Thekkumkara Surendran Anish, Jerin Jose Cherian, Mahendra Pratap Singh, Ashish Kumar Kakkar, Aparna Mukherjee, Devi Vijay.

**Project administration:** Rachna Rohilla, Mayank Gupta, Mahendra Pratap Singh, Ashish Kumar Kakkar, Niti Mittal, Sandeep Kaushal, Devi Vijay, Robin Kaushik, Syed Shariq Naeem, Jaykaran Charan.

**Resources:** Rachna Rohilla.

**Supervision:** Aparna Mukherjee, Rachna Rohilla, Mayank Gupta, Thekkumkara Surendran Anish, Jerin Jose Cherian, Mahendra Pratap Singh, Ashish Kumar Kakkar, Niti Mittal, Sandeep Kaushal, Devi Vijay, Robin Kaushik, Syed Shariq Naeem, Jaykaran Charan.

**Validation:** Rachna Rohilla, Mayank Gupta, Mahendra Pratap Singh, Ashish Kumar Kakkar, Niti Mittal.

**Writing – original draft:** Rachna Rohilla, Mayank Gupta, Thekkumkara Surendran Anish, Mahendra Pratap Singh, Aparna Mukherjee.

**Writing – review & editing:** Rachna Rohilla, Mayank Gupta, Thekkumkara Surendran Anish, Jerin Jose Cherian, Mahendra Pratap Singh, Ashish Kumar Kakkar, Aparna Mukherjee, Niti Mittal, Sandeep Kaushal, Devi Vijay, Robin Kaushik, Syed Shariq Naeem, Jaykaran Charan.

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
