## [Editor Report · Decision Letter 0]

6 Feb 2025

MULTIPRONGED INTERVENTIONS TO REDUCE SURGICAL SITE INFECTIONS: A MULTICENTER IMPLEMENTATION RESEARCH PROTOCOL

PONE-D-24-54630

Dear Dr. Rohilla,

As a Peer-Reviewed Funded Protocol, your submission is eligible for expedited review by in-house editors. Based on our evaluation, we are satisfied that your manuscript meets our publication criteria for Study Protocols, and is therefore considered to be suitable for publication subject to final journal requirements.

Kind regards,

Steve Zimmerman, PhD

Senior Editor, PLOS One

1.  Thank you for stating the following financial disclosure:

“This study will be supported by funding from the Indian Council of Medical Research under NTF-SRUM file number. ICMR/EM/SRUM/2023/Rachna (E-office No. 171264) dated 06/03/2024”

Please respond by return e-mail so that we can amend your financial disclosure and competing interests on your behalf.

“The funder has appointed the study monitor [AK, Clinical Studies and Trials Unit, Division of Development Research, ICMR, New Delhi], under whose guidance the research will be conducted, in coordination with [JJC, Clinical Studies and Trials Unit, Division of Development Research, ICMR, New Delhi]. All other authors: none to declare.”

3. One of the noted authors is a group or consortium [IMPRESS (‘Impact of Multi-Pronged intervention on REducing Surgical Site Infection’) Study Group]. In addition to naming the author group, please list the individual authors and affiliations within this group in the acknowledgments section of your manuscript. Please also indicate clearly a lead author for this group along with a contact email address.

6. We note that Figure 4 in your submission contain [map/satellite] images which may be copyrighted. All PLOS content is published under the Creative Commons Attribution License (CC BY 4.0), which means that the manuscript, images, and Supporting Information files will be freely available online, and any third party is permitted to access, download, copy, distribute, and use these materials in any way, even commercially, with proper attribution. For these reasons, we cannot publish previously copyrighted maps or satellite images created using proprietary data, such as Google software (Google Maps, Street View, and Earth). For more information, see our copyright guidelines: http://journals.plos.org/plosone/s/licenses-and-copyright.

a. You may seek permission from the original copyright holder of Figure 4 to publish the content specifically under the CC BY 4.0 license. 

7. We note that the original protocol file you uploaded contains a confidentiality notice indicating that the protocol may not be shared publicly or be published. Please note, however, that the PLOS Editorial Policy requires that the original protocol be published alongside your manuscript in the event of acceptance. Please note that should your paper be accepted, all content including the protocol will be published under the Creative Commons Attribution (CC BY) 4.0 license, which means that it will be freely available online, and any third party is permitted to access, download, copy, distribute, and use these materials in any way, even commercially, with proper attribution.

Therefore, we ask that you please seek permission from the study sponsor or body imposing the restriction on sharing this document to publish this protocol under CC BY 4.0 if your work is accepted. We kindly ask that you upload a formal statement signed by an institutional representative clarifying whether you will be able to comply with this policy. Additionally, please upload a clean copy of the protocol with the confidentiality notice (and any copyrighted institutional logos or signatures) removed.
---

## [Editor Report · Acceptance letter]

PONE-D-24-54630

PLOS ONE

Dear Dr. Rohilla,

I'm pleased to inform you that your manuscript has been deemed suitable for publication in PLOS ONE. Congratulations! Your manuscript is now being handed over to our production team.

Kind regards,

on behalf of

Dr Steve Zimmerman

Staff Editor

PLOS ONE